# Multiple parallel origins of parasitic Marine Alveolates

Corey C. Holt [1,2,7] ✉, Elisabeth Hehenberger [1,3,7] ✉, Denis V. Tikhonenkov[4,5], Victoria K. L. Jacko-Reynolds [1], Noriko Okamoto [1,2], Elizabeth C. Cooney [1,2], Nicholas A. T. Irwin [1,6] & Patrick J. Keeling [1] ✉

Microbial eukaryotes are important components of marine ecosystems, and the Marine Alveolates (MALVs) are consistently both abundant and diverse in global environmental sequencing surveys. MALVs are dinoflagellates that are thought to be parasites of other protists and animals, but the lack of data beyond ribosomal RNA gene sequences from all but a few described species means much of their biology and evolution remain unknown. Using single-cell transcriptomes from several MALVs and their free-living relatives, we show that MALVs evolved independently from two distinct, free-living ancestors and that their parasitism evolved in parallel. Phylogenomics shows one subgroup (MALV-II and -IV, or Syndiniales) is related to a novel lineage of free-living, eukaryovorous predators, the eleftherids, while the other (MALV-I, or Ichthyodinida) is related to the free-living predator *Oxyrrhis* and retains proteins targeted to a non-photosynthetic plastid. Reconstructing the evolution of photosynthesis, plastids, and parasitism in early-diverging dinoflagellates shows a number of parallels with the evolution of their apicomplexan sisters. In both groups, similar forms of parasitism evolved multiple times and photosynthesis was lost many times. By contrast, complete loss of the plastid organelle is infrequent and, when this does happen, leaves no residual genes.

The marine alveolates (MALVs) are an elusive group of microbial eukaryotes first discovered through amplicon surveys of environmental rRNA[1,2], and since found to consistently dominate eukaryotic metabarcoding surveys in global oceans[3–8]. Virtually all MALVs are known only from these rRNA SSU gene fragments, which have been phylogenetically linked to a handful of described parasites that branch deeply in the dinoflagellates[1,9–11]. MALV sequences in marine environments are accordingly interpreted to represent a large and diverse population of dinospores that infect animal or protist hosts. Their abundance, diversity, and phylogenetic position all emphasize the evolutionary and ecological significance of MALVs but, since their

discovery, new insights beyond their SSU rRNA gene diversity have been limited by a lack of genomic data and the inability to culture all but a few species.

MALVs are typically subdivided into five groups based on rRNA (I-V) and are often considered to be a monophyletic clade comprising the order Syndiniales[7]. Phylogenetic analyses support the branching of MALVs at the base of the dinoflagellates, but concatenating SSU and LSU rRNA genes suggests MALVs may be paraphyletic, with MALV-II and -IV forming one group and MALV-I diverging earlier[12]. Comprehensive genomic data are only available for two described MALVs, *Amoebophrya* and *Hematodinium*, which are in the MALV-II and IV

[1]Department of Botany, University of British Columbia, Vancouver, British Columbia, Canada. [2]Hakai Institute, Heriot Bay, British Columbia, Canada. [3]Institute of Parasitology, Biology Centre Czech Academy of Sciences, České Budějovice, Czech Republic. [4]Papanin Institute for Biology of Inland Waters, Russian Academy of Sciences, Borok, Russia. [5]AquaBioSafe Laboratory, University of Tyumen, Tyumen, Russia. [6]Present address: Merton College, University of Oxford, Oxford, UK. [7]These authors contributed equally: Corey C. Holt, Elisabeth Hehenberger. ✉e-mail: corey.holt@ubc.ca; elisabeth.hehenberger@paru.cas.cz; pkeeling@mail.ubc.ca

groups, respectively[13–15]. Both lack evidence of a plastid, despite the fact that plastids are found in all other closely-related dinoflagellate lineages[14,15]. Losing photosynthesis is relatively common, but losing the plastid entirely is not[14,16], and this absence of a plastid has created some controversy about the evolution of the organelle in close relatives[17]. However, the lack of data beyond rRNA gene sequencing for most MALVs, and in particular the major MALV-I group (with the exception of uncultured Single-cell Assembled Genome (SAG) data), together with the lack of data from free-living relatives of these apparently obligate parasites, makes it difficult to untangle the evolution of parasitism and plastids.

Here, we show, using single-cell transcriptomes and phylogenomic analysis, that parasitism evolved multiple times in MALVs from two distinct, free-living lineages. MALV-II and MALV-IV (the Syndiniales) share a common ancestor with a new group of free-living heterotrophs, the eleftherids, whereas MALV-I shares a common ancestor with a different free-living lineage, *Oxyrrhis marina*. Transcriptomic data from the psammosids confirm their position as early-branching dinoflagellates, while new MALV-I data support the need to revise the genus *Euduboscquella* and suggest MALV-I (now the Ichthyodinida) retains a cryptic plastid and should no longer be considered part of the Syndiniales.

## Results and discussion

### MALV-II and -IV are closely related to the eleftherids, a new group of free-living heterotrophs

We isolated and cultured several strains collectively representing two new species of colorless, eukaryovorous flagellates, which we formally name *Eleftheros xomoi* (strains Colp-37 and Cur-11, isolated from the surface of a coral in the Caribbean Sea, Curaçao) and *Eleftheros karadeniz* (strain Colp-25, isolated from marine near-shore sediments in the Black Sea, Crimea) (see "Taxonomic summary" below for taxonomic diagnosis). Sequences sufficiently similar to eleftherid SSU rRNA genes were not found in any of the largest global planktonic environmental surveys[18,19], but we did recover small numbers of amplicons from marine sediment data, with a higher prevalence (albeit with often low read counts) found in deep sea samples[20] (Supplementary Fig. 1; Supplementary Data 1), suggesting that eleftherids are rare and benthic. *Eleftheros* are very small (~4 μm), bean-shaped or roundish, fast-swimming cells with two heterodynamic flagella, externally resembling the biflagellate dinospores of MALVs (Fig. 1, Supplementary Fig. 2 and Supplementary Movie 1). Many of their morphological features are reminiscent of dinoflagellates, including longitudinal and transverse grooves similar to cingulum and sulcus (Fig. 1a, c, f, j–m, o, q). Eleftherids possess large alveoli beneath the plasma membrane (Fig. 1n–r), flagellar basal bodies orientated predominantly at an acute angle to each other, a flagellar transition zone including an axosome with extended central microtubules and a transverse plate just below the cell surface (Fig. 1p and Supplementary Fig. 2a), enlarged perinuclear space containing tubular mastigonemes (Fig. 1n, q and Supplementary Fig. 2b), bowling pin-shaped trichocysts with square cross-sections scattered throughout the cytoplasm that are cross-striated after discharging (Fig. 1n, o, s and Supplementary Fig. 2c–e), a large convoluted mitochondrion with tubular cristae (Fig. 1n, o, q and Supplementary Fig. 2a, f), and storage compounds in the form of roundish granules (Fig. 1n and Supplementary Fig. 2a, g). Interestingly, we also observed large and distinctive vesicular compartments resembling the rhoptries of apicomplexan and perkinsid parasites, which are also present in the parasitic MALVs (Fig. 1o, r).

To determine their phylogenetic position, we generated transcriptomic data from both whole cultures and manually isolated eleftherid cells and built a concatenated 75-taxon/236 protein dataset (63,267 sites). Maximum-likelihood (ML) analyses consistently recovered eleftherids as sister to *Amoebophrya* and *Hematodinium* (MALV-II and -IV) with maximum statistical support (Fig. 2a). To examine the

possibility that long-branch attraction affected the position of eleftherids, we performed fast-evolving site removal analysis until 50% of sites were removed, which showed that the support for the eleftherids as sisters to MALV-II and -IV remained 100% (Fig. 2b).

### MALV-I is closely related to *Oxyrrhis*, another free-living heterotroph

Currently, only rRNA gene fragments and a small number of SAGs are available from the other major MALV group, MALV-I, and none of this group are in culture. Accordingly, we sought infected hosts from marine samples and obtained one infected cell from a deep-water plankton tow near Quadra Island, British Columbia, and two cells from near-shore plankton tows at Piscadera Bay, Curaçao. The Quadra Island cell was identified from its morphology as the dinoflagellate *Polykrikos* sp. but was also observed to contain a large and distinctive circular "fried egg-like" inclusion measuring ~75 μm by 82 μm with a central nucleus and centripetal grooves on a cuticular disc (Fig. 2c; Supplementary Movie 2); a common morphological characteristic of a mature trophonts in MALV-I parasites, including others infecting dinoflagellates[21]. Of the two cells from Curaçao: one was an infected *Warnowia* sp. with a smaller, ovoid, brown inclusion measuring ~27 μm by 35 μm (Fig. 2d; Supplementary Movie 3), and the other was found with no discernable host. While the morphology of the former lacks sufficient resolution for definitive descriptions, it appears to contain a "brain-like" groove texture (Supplementary Movie 3) reminiscent of *Euduboscquella* sp. 4 (OP445725) and ciliate-infecting *Euduboscquella* species[21]. The remaining cell from Curaçao, on the other hand, showed clear centripetal grooves and ridges, measured ~68 μm by 70 μm, and resembled a late-stage trophont or perhaps a newly emerged tomont (Fig. 2e; Supplementary Movie 4). Although we could not determine the number of grooves (which can distinguish species of *Euduboscquella*) in each cell, like those described in Yoo et al.[21], cells with this morphology appeared to be more similar (in terms of number of grooves) to *E. melo* (~30) compared to *E. nucleocola* (more than 100)[22], both of which are dinoflagellate-infecting *Euduboscquella* spp.

All three cells were isolated, and single-cell transcriptomes were sequenced, which revealed two phylogenetically-distinct signals in two of the cells—one being the above-mentioned hosts and the other an uncharacterized MALV-I (Fig. 2a)—while no potential host sequences were found in the remaining cell. Phylogenetic analysis of full-length SSU rRNA gene sequences extracted from the *Polykrikos*-associated transcriptomic data showed the MALV-I parasite (herein referred to as Ichthyodinida sp. 1) clustering with full-support with a clade of unidentified environmental sequences in the MALV-I clade (Fig. 3). This fell within a larger group of parasites isolated from dinoflagellates, which were ascribed to *Euduboscquella* based on conserved morphological traits[21]. This group corresponds to MALV-I clade 3 elsewhere[7]. The MALV-I sequence derived from *Warnowia*-associated transcriptomic data (herein referred to as Ichthyodinida sp. 2), on the other hand, branched at the base of the ciliate-infecting *Euduboscquella* clade 4[7], while the remaining, apparently host-less, MALV-I cell (Ichthyodinida sp. 3) clustered within a group of environmental sequences (MALV-I clade 5) noted to be one of the most commonly retrieved groups in environmental libraries[7]. *Euduboscquella* has been isolated from both ciliates and dinoflagellates[21–25]; however, the SSU rRNA gene tree (Fig. 3; Supplementary Fig. 3) showed MALV sequences isolated from these two host types form two phylogenetically discrete groups. Indeed, MALV-I diversity seems to fall into a small number of highly host-specific clades, infecting ciliates, fish eggs, dinoflagellates, and radiolarians. However, host specificity is noted to vary between MALV lineages, with some infecting a single host species and others infecting multiple[25–27]. That being said, the branch lengths of MALV-I lineages in phylogenomic analyses (Fig. 2a; Supplementary Fig. 4), together with the presence of multiple distinct *Euduboscquella* clades in SSU phylogenies (Fig. 3; Supplementary Fig. 5), all suggest that the genus

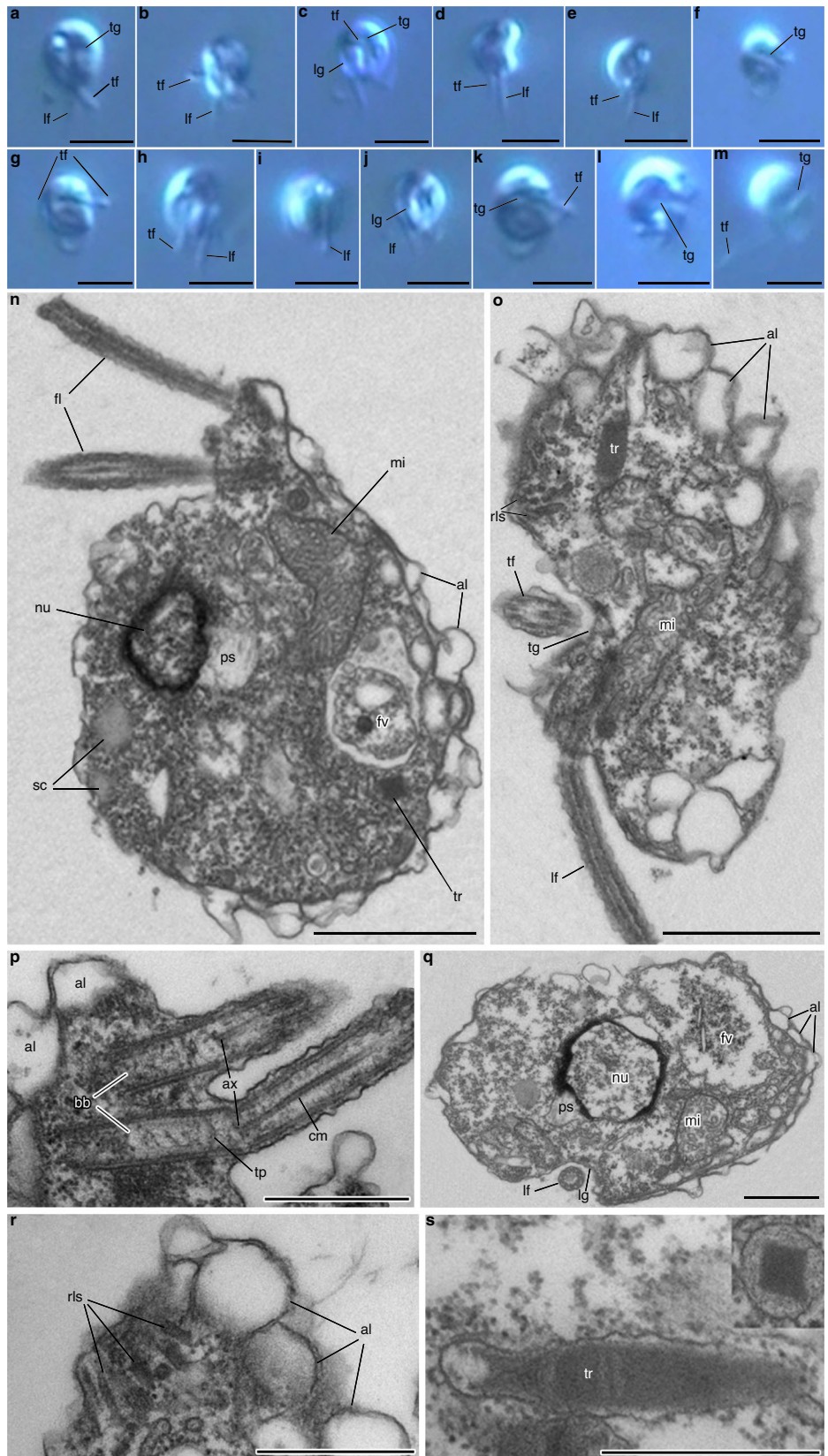

**Fig. 1 | Cell morphology of *Eleftheros*. a–j** Living cells of *E. xomoi* (strain Cur-11), visualized by light microscopy. **k–m** Living cells of *E. karadeniz*, visualized by light microscopy. **n–s** Cells of *E. karadeniz*, visualized by transmission electron microscopy. **n** Longitudinal section. **o** Oblique section showing the transverse groove. **p** Arrangement of basal bodies. **q** Transverse section showing the longitudinal groove. **r**,Structure of the distal part with rhoptry-like structures and alveoli. **s** Longitudinal and cross section (inset) of trichocysts. al, alveoli; ax, axosome; bb, basal bodies; cm, central microtubules; fl, flagella; fv, food vacuole; lf, longitudinal flagellum; lg, longitudinal groove; mi, mitochondrion; nu, nucleus; ps, perinuclear space; rls, rhoptry-like structures; sc, storage compounds; tr, trichocyst; tf, transverse flagellum; tg, transverse groove; tp, transverse plate. Scale bars, 4 µm (**a–m**), 1 µm (**n**, **o**, **q**), and 0,5 µm (**p**, **r**, **s**). All observations were repeated at least 3 times with similar results.

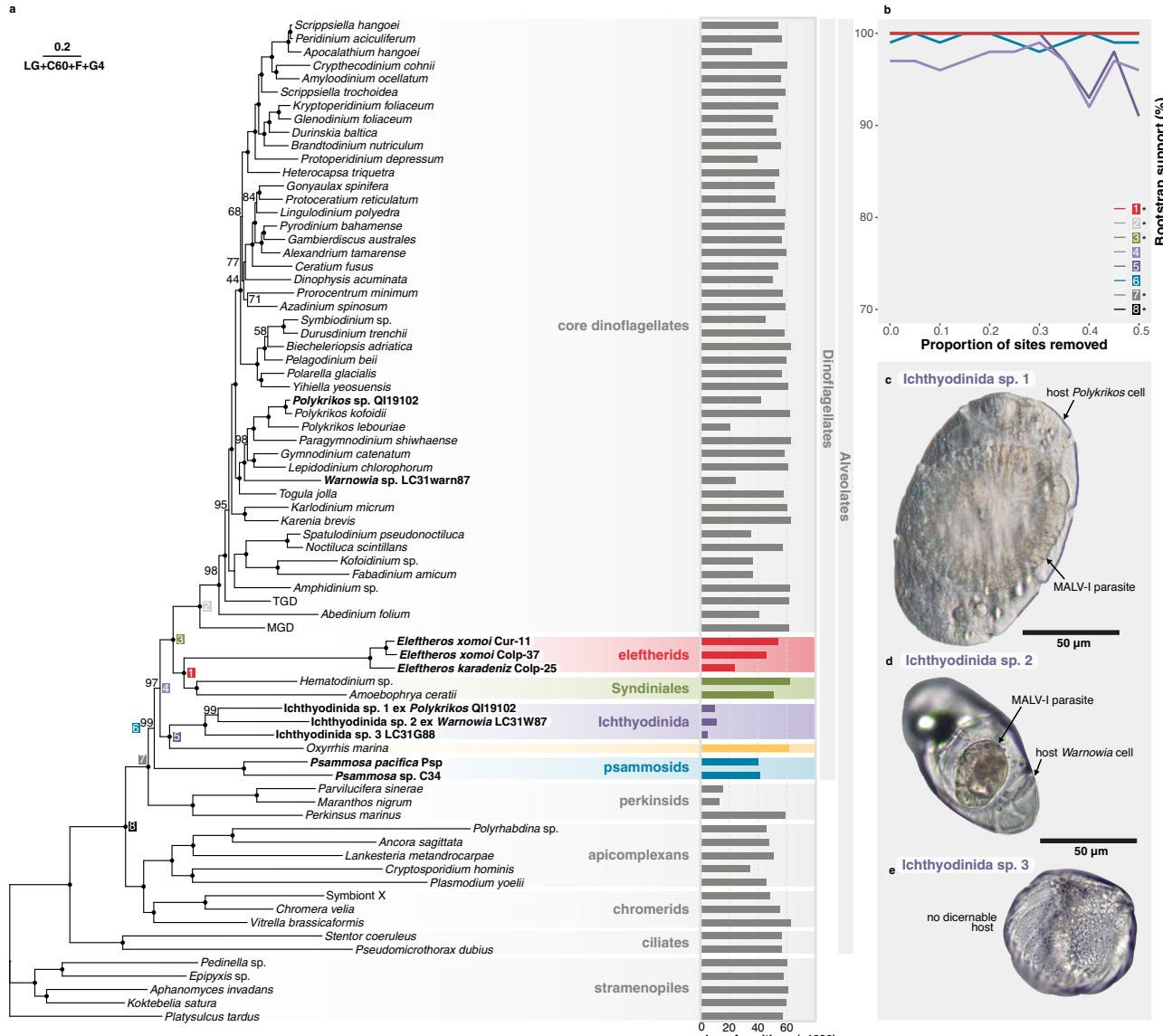

**Fig. 2 | Phylogenomic reconstruction of the Marine Alveolates. a** ML analysis of a 236 multi-protein alignment (LG + C60 + F + G4) separates MALV-I (purple) from the remaining Syndiniales (MALV-II and IV, green) to which the eleftherids (red) are the free-living sister group. Psammosids are shown in blue and *Oxyrrhis marina* in yellow. Black dots represent full statistical support (non-parametric bootstrap = 100%), values are shown for support below 100%. The percentage of amino acids present in the corresponding alignment for each taxon are shown as bars to the right of the tree. **b** UltraFast bootstrap support for given topologies, indicated by highlighted node labels in (**a**), following removal of the fastest-evolving sites as inferred using the LG + C 40 + F + G4 model. * indicates nodes that remained at 100%. Support for core dinoflagellate monophyly serves as a control for the presence of sufficient information for phylogenomic inference. **c** Living cell of *Polykrikos* sp. infected with a MALV-I-like parasite (Ichthyodinida sp. 1) showing characteristic centripetal grooves. **d** Living cell of *Warnowia* sp. infected with a MALV-I-like parasite (Ichthyodinida sp. 2). **e** MALV-I cell (Ichthyodinida sp. 3) showing no discernable host but characteristic centripetal grooves. Scale bar, 50 μm.

*Euduboscquella* requires revision, with two or potentially three additional genera needed to reflect the molecular diversity shown here.

Phylogenomic analyses including these new data reject the monophyly of MALV-I and MALV-II/IV, instead showing complete support for MALV-I branching specifically with another lineage of free-living heterotrophs, the common marine flagellate *Oxyrrhis* (Fig. 2a). This topology is further supported by the addition of published SAG data[28] (Supplementary Fig. 4). All topologies in which MALVs formed a monophyletic group were rejected by approximately unbiased (AU) tests (Supplementary Fig. 6). Indeed, MALVs, as currently defined, are neither monophyletic nor paraphyletic, since we can demonstrate complete support for separate lineages being sister groups to different free-living heterotrophs. Parasitic transitions in MALV-I and MALV-II therefore trace back to different free-living ancestors and must have evolved their obligate parasitic life independently. We propose to retain the name Syndiniales for MALV-II/IV and to use the order Ichthyodinida proposed by Cavelier-Smith for MALV-I[29] (as opposed to the Coccidiniales, which has been previously proposed but is based on a problematic type, *Coccidinium*[23,30,31]). Whether MALV-III and the little-studied MALV-V can all be included in Ichthyodinida remains to be seen[7] (Fig. 3, Supplementary Fig. 3).

**Plastids and parasitism**

The complete absence of evidence for a plastid in *Amoebophrya* and *Hematodinium*[14,15] led to new hypotheses to account for the apparently complex distribution of plastids in early-diverging dinoflagellates and their sister clades, the apicomplexa, chrompodellids, and squirmids (or ACS clade)[32–34]. However, it is difficult to evaluate any hypothesis

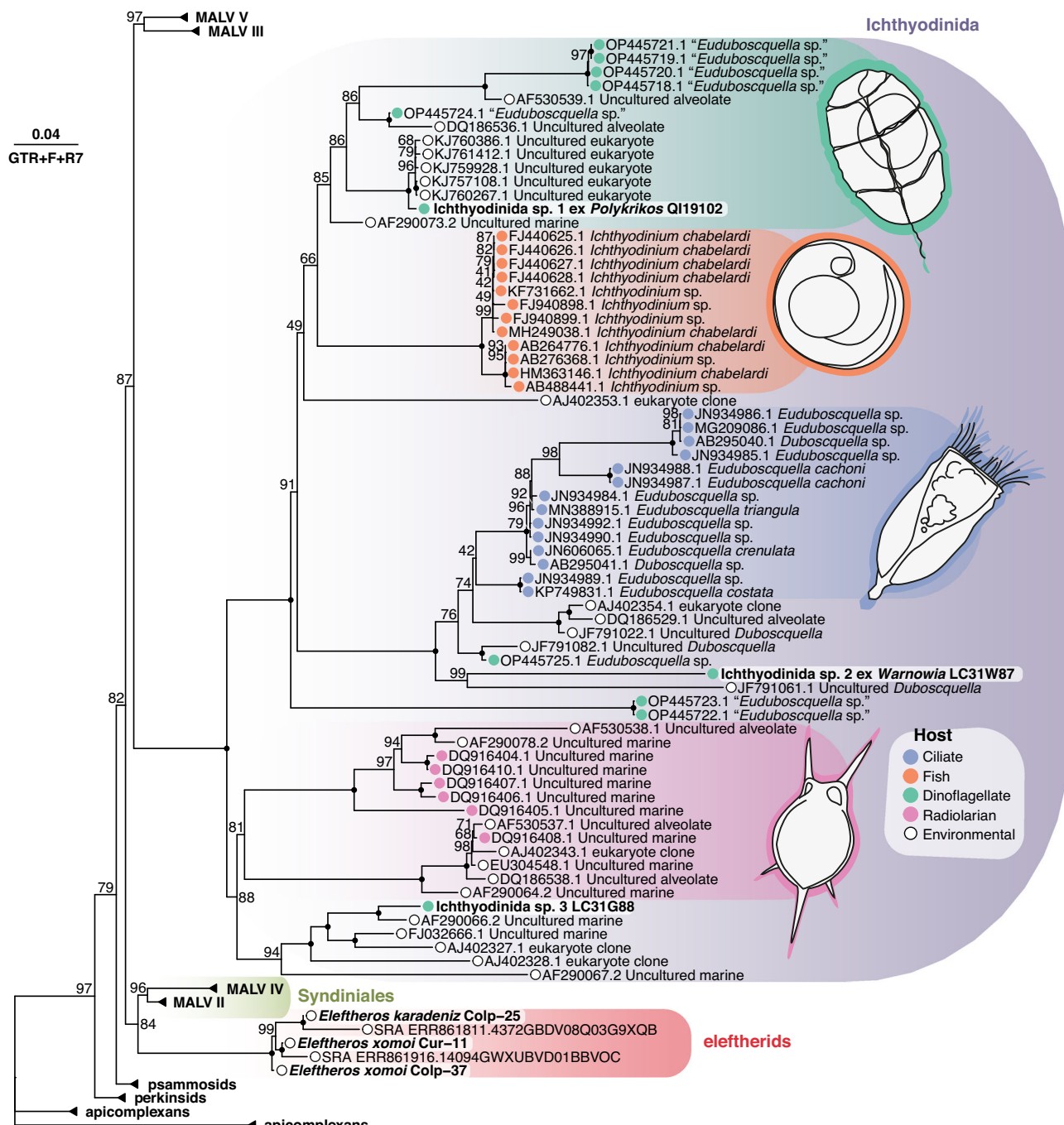

**Fig. 3 | Contextualising new MALVs using SSU data.** ML analysis (GTR + F + R7) showing MALV-I lineages (purple) cluster by host and those isolated in this study belong to distinct MALV-I clades. The eleftherids (red) again form a sister clade to the remaining Syndiniales (MALV-II and MALV-IV, green). Black dots at nodes represent full statistical support (UltraFast bootstrap = 100%), values are shown for support below 100%. Colored circles accompanying MALV-I lineages reflect host identity: ciliates, blue; fish, orange; dinoflagellates, green; radiolarians, pink; and environmental, white. *Oxyrrhis* is not included due to its highly divergent rRNA SSU gene sequences.

about the origin of plastids in parasites without data from close free-living relatives. Now that such relatives are identified, we examined plastid function across the deeply-branching dinoflagellate groups, including another relevant lineage, *Psammosa*. *Psammosa* is a free-living heterotrophic dinoflagellate, but its exact position and whether it has a plastid are unclear in the absence of genomic data[35]. *Psammosa pacifica* appears relatively rare in marine sediment datasets (Supplementary Data 1) but we re-established a culture of *P. pacifica* from the same location as the original description, alongside that of a novel lineage (with a partial SSU sequence 88.91% identical to *P. pacifica*),

and generated two transcriptomes from both strains. Phylogenomic analyses strongly support them forming an independent lineage at an important juncture in the tree, although exactly where they branch remains uncertain. In most analyses they branch basal to *Oxyrrhis*, Ichthyodinida, Syndiniales, and core dinoflagellates (Fig. 2a), but a position within the *Oxyrrhis*/Ichthyodinida clade itself is not rejected (Supplementary Fig. 6).

Eleftherids and (to a lesser extent) MALV-I and psammosids all possessed substantial numbers of nucleus-encoded genes for plastid-targeted proteins (as does *Oxyrrhis*), and these relate to metabolic

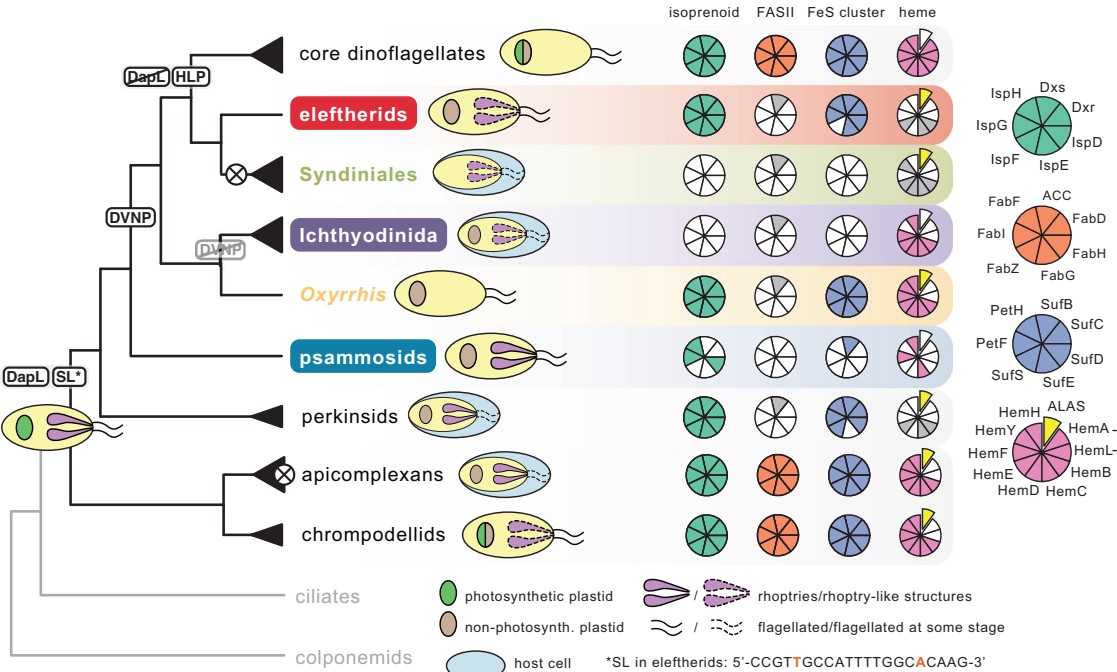

**Fig. 4 | Character and plastid metabolic pathway evolution in dinoflagellates, apicomplexans, and related lineages.** Schematic tree indicating abundance of and character evolution in dinoflagellates, apicomplexans, and related lineages. Black triangles/lines denote an estimation of the relative diversity of each lineage, while crossed-out circles indicate plastid loss (i.e., in the ancestor of the MALVs and within the apicomplexans). Taxa colored according to Fig. 2a: eleftherids, red; Syndiniales (MALV-II and -IV), green; Ichthyodinida (MALV-I), purple; *Oxyrrhis marina*, yellow; and psammosids, blue. Abbreviations on branches indicate the inferred origin of: L,L-diaminopimelate aminotransferase, DapL; SL, spliced leader; DVNPs, Dinoflagellate Viral NucleoProteins; HLPs, histone-like proteins. Next to each branch, Coulson plots depict selected plastidial metabolic pathways in the corresponding lineage. Grey segments indicate a cytosolic form of a given protein, while an offset, yellow segment denotes the mitochondrial enzyme 5-aminolevulinate synthase (ALAS) that synthesizes 5-aminolevulinate in all lineages but the core dinoflagellates, where the same intermediate is synthesized by the plastidial enzymes glutamyl-tRNA reductase (HemA) and glutamate-1-semialdehyde 2,1-aminomutase (HemL). Red nucleotides in SL sequence indicate deviation from canonical SL. Pale blue ovals surrounding yellow cell depictions show host cells, indicating parasitism. Coulson plot headers refer to metabolic pathways: isoprenoid, methylerythritol phosphate (MEP) pathway for isoprenoid biosynthesis (green segments); FASII, type 2 fatty acid biosynthesis pathway (orange segments); FeS cluster, plastidial Fe-S cluster biosynthesis pathway including a ferredoxin system (navy blue segments); heme, heme biosynthesis (pink segments). Complete protein names are listed in Supplementary Data 2.

pathways typical for non-photosynthetic plastids[36]. Specifically, in eleftherids and psammosids, we identified genes for the methylerythritol phosphate (MEP) pathway for isoprenoid biosynthesis and a plastidial SUF-type FeS cluster biosynthesis pathway including a ferredoxin redox system (Fig. 4 and Supplementary Fig. 7; phylogenetic trees available at https://doi.org/10.6084/m9.figshare.19351700). These are both widespread in non-photosynthetic dinoflagellates (non-photosynthetic core dinoflagellates and *Oxyrrhis*)[36,37] and apicomplexans[34] but absent in Syndiniales[14]. The complete transcripts encode N-terminal extensions with one or more predicted transmembrane domains that are consistent with plastid targeting (IspF, SufC, SufD, SufE, PetF, PetH—in eleftherids), but do differ from canonical plastid-targeted sequences in core dinoflagellates[38] (alignments available at https://doi.org/10.6084/m9.figshare.19351700). No evidence for the plastidial type II fatty acid biosynthesis pathway was found, which is also absent in MALVs and all other dinoflagellates with non-photosynthetic plastids[36,39] (Fig. 4 and Supplementary Fig. 7). Fatty acid biosynthesis in these groups is thought to occur via a cytosolic type I fatty acid synthase[36] (Supplementary Data 2). Heme is also synthesized in the plastids of some apicomplexans and dinoflagellates, but in eleftherids all these enzymes were mitochondrial (5-aminolevulinate synthase ALAS) or cytosolic (porphobilinogen synthase HemB, hydroxymethylbilane synthase HemC, uroporphyrinogen decarboxylase HemE), which is also the case in Syndiniales and *Perkinsus*. *Oxyrrhis*, by contrast, retains a mixed pathway with some enzymes inferred to be in the plastid, like dinoflagellates and apicomplexans (Supplementary Fig. 7; phylogenetic trees available at

https://doi.org/10.6084/m9.figshare.19351700). Like *Oxyrrhis*, we found MALV-I genomes encode HemB, HemC, HemE, HemF, and HemH which are all inferred to be targeted to the plastid: HemC exhibits a clear transmembrane region (TMR) which transports plastid proteins through the Golgi[38]; HemB, HemH and HemF are truncated at the N-terminus but cluster with plastid-targeting dinoflagellates (like *Oxyrrhis*) in phylogenetic analyses; while HemE also contains N-terminal targeting information and falls within the so-called "eukaryotic" clade containing aplastidial organisms and some plastid-targeting dinoflagellates like *Oxyrrhis*. Notably HemE sequences from *Perkinsus*, *Hematodinium* and *Amoebophrya* do not have N-teminal extensions and are likely cytosolic. HemD (uroporphyrinogen-III synthase), the only enzyme encoded by MALV-II genomes that was proposed to have originated in the plastid[14], is truncated in MALV-I. We did not find a HemD orthologue in eleftherids or psammosids (Fig. 4 and Supplementary Fig. 7), and it is also absent from many dinoflagellate transcriptomes, suggesting a low level of expression. However, in our phylogenetic reconstruction of HemD (Supplementary Fig. 8), photosynthetic eukaryotes and MALVs do not group with cyanobacteria, suggesting a bacterial origin for HemD independent from the plastid. The retention of plastidial versions of enzymes involved in heme biosynthesis in both MALV-I and *Oxyrrhis*, together with their absence in eleftherids and Syndiniales (Supplementary Fig. 7), indicates a functional redundancy that persisted long into the evolution of dinoflagellates and may have facilitated the repeated loss of plastid heme biosynthesis, and perhaps even the loss of the plastid itself in MALV-II.

The genome of *Hematodinium* (MALV-IV) contains another potential plastid pathway, the lysine biosynthesis enzyme L,L-diaminopimelate aminotransferase (DapL), which is only present in bacteria and a subset of plastids. Its phylogenetic origin in *Hematodinium* could not be resolved and support for its plastid origin was concluded to be ambiguous[14]. We identified DapL in eleftherids and psammosids (Fig. 4; protein absent in apicomplexans and most dinoflagellates) and phylogenetic reconstruction shows it to group with plastid homologs in cryptophytes, whereas the *Hematodinium* homolog appears to have originated independently from bacteria (Supplementary Fig. 9). While most eukaryotic DapL genes encode plastid-targeted sequences, eleftherid and cryptophyte genes do not, indicating a likely cytosolic function (alignment available at https://doi.org/10.6084/m9.figshare.19351700). We could not identify DapL sequences in MALV-I.

We found no proteins related to photosynthesis or chlorophyll biosynthesis in eleftherids, psammosids, MALV-I, or *Oxyrrhis* (as expected). We also identified no plastid import proteins; however, the import machinery is understudied in dinoflagellates and only a single component has been reported[40]. There is no evidence for a plastid genome in any of the taxa investigated here, which is also expected since the plastid genome of photosynthetic dinoflagellates encodes only 12 proteins that are exclusively involved in photosynthesis[41]. We also investigated the 129 proteins predicted to localize to the apicoplast in the apicomplexan *Toxoplasma* by spatial proteomics[42] and found no clear candidates for plastid-targeted homologs beyond the plastid pathways noted above. Taken together, the plastid in these non-photosynthetic lineages has been profoundly reduced.

## Early character evolution in dinoflagellate-related lineages

Dinoflagellates evolved in many strange ways, beyond parasitism and plastids[36,43,44], and we now have a more accurate glimpse into these events as well. The nucleus, or "dinokaryon" is a good example: unlike other eukaryotes, dinoflagellates lack bulk nucleosomal chromatin and have huge genomes with permanently-condensed chromosomes structured by some combination of genes derived from horizontal gene transfer from viruses (Dinoflagellate Viral NucleoProtein, or DVNP)[45,46] and bacteria (two Histone-Like Proteins, or HLPs). DVNP is present in all taxa analyzed here, with the notable exceptions of both psammosids and all MALV-I (Fig. 4 and Supplementary Data 2), whereas HLPs are present only in core dinoflagellates[36]. DVNP is highly expressed and therefore generally easy to detect in dinoflagellate transcriptomes, so its complete absence from psammosid and MALV-I datasets raises the possibility that the gene might not be present. This is consistent with the phylogenetic position of *Psammosa*, but would require MALV-I to have lost this critical gene. Genomic data and/or biochemical evidence are needed to confirm this.

Dinoflagellate nuclei also perform trans-splicing of a highly-conserved spliced leader (SL) to the 5' end of many or most mRNAs[47]. The same is also true of the perkinsids, however, the SL sequence is more variable[48]. We identified complete SL sequences in a subset of transcripts from eleftherids, MALV-I, and psammosids, and found eleftherids showed variations in sequence from the canonical SL sequence identified in core dinoflagellates and MALVs[15,47] (Fig. 4).

The ultrastructure of eleftherids also contains links to MALVs, which shed light on their diversification. Electron microscopy revealed the presence of rhoptry-like structures (Fig. 1o, r), similar to the electron-dense bodies found in the MALV-II *Amoebophrya*[49] and MALV-I *Ichthyodinium*[10]. Rhoptries are components of the infection apparatus of apicomplexan parasites, their free-living relatives, as well as *Perkinsus* and *Psammosa*[50]. In eleftherids the presence of rhoptry-like structures may indicate components of an apical complex were also retained with a role in feeding, as is the case in *Psammosa* and a handful of free-living lineages related to apicomplexans[51]. Altogether the distribution of rhoptry-like structures and plastids supports a mixotrophic ancestor of dinoflagellates and apicomplexans (Fig. 4).

## Parallelism in the evolution of parasitism

While we have focused mostly on the parasitic dinoflagellates, of course, the sisters to dinoflagellates are the much more famously parasitic apicomplexans (e.g., the malaria parasite *Plasmodium*). Due to their medical significance, apicomplexans are far better studied, and since it became clear they too evolved from algae and retained a cryptic plastid, their evolution has been thoroughly investigated[52,53]. In the ACS clade, it is now clear that many of the major transitions seemingly associated with the origin of parasitism actually evolved multiple times in parallel. This includes the loss of photosynthesis, the loss of the plastid organelle, and even the origin of parasitism itself[54]. Marine environmental surveys show MALVs dominate eukaryotic communities in terms of both diversity and abundance. Accordingly, they are likely to play an important role in marine ecology, yet we know shockingly little about their biology or evolution. It is now at least becoming clear that some of the same patterns emerging from apicomplexan research are also true of parasitic dinoflagellates, in particular parallel evolution in seemingly similar parasites.

Syndiniales (MALV-II and -IV) and Ichthyodinida (MALV-I) were previously thought to have evolved from a common parasitic ancestor[1,2,7], but phylogenomics now shows that they too evolved independently from different free-living ancestors, or in other words, that parasitism evolved twice in the Marine Alveolates. This is based on the same reasoning that is commonly applied to the ACS clade, namely that a transition from a free-living state to an obligately intracellular and parasitic state is more likely than a reversion from an obligate intracellular parasite to a free-living form. This is further supported by the fact that adaptation to intracellular parasitism is often reductive, and it is hard to reinvent complex traits (for example, feeding by phagocytosis) once lost. Moreover, because plastids are present in nearly every apicomplexan and dinoflagellate subgroup, and are demonstrably related[54], the ancestor of all these groups can be inferred to also have had a plastid. And because photosynthesis is still found in both dinoflagellates and ACS taxa (and they share a form of Rubisco unique among eukaryotes), this ancestor can also be inferred to have been photosynthetic. The prevalence of non-photosynthetic organisms branching at the base of the dinoflagellate tree has led to complex theories about the origin of their plastids[17]. However, if you also consider the ACS clade, a relatively straightforward model with multiple parallel losses of photosynthesis in early dinoflagellate evolution may not be that remarkable. While it is true that many cryptic plastids are found in early-branching dinoflagellates (MGD, *Abedinium*, TGD, *Eleftheros*, *Psammosa* and *Oxyrrhis*) and *Perkinsus*, they are all serviced by nucleus-encoded, plastid-targeted proteins that are phylogenetically related to homologs in peridinin plastids of photosynthetic dinoflagellates (https://doi.org/10.6084/m9.figshare.19351700), which very strongly argues they all evolved from the same ancestor. Based on the current tree, this would require multiple independent losses of photosynthesis, and it is these parallel events that have prompted more complex theories to explain plastid distribution[17]. However, if you count the number of losses in deep-branching taxa on the ACS side of the tree, you arrive at roughly the same number of independent losses of photosynthesis. Yet, multiple losses in the ACS clade attract no serious debate and are widely accepted to be the result of convergence. The key difference appears to be that the large, diverse, ecologically-successful apicomplexans are non-photosynthetic, so the retention of photosynthesis in a few "deep-branching" lineages (*Chromera* and *Vitrella*) leads to those lineages being treated as "oddballs". On the dinoflagellate side of the tree, it is the large and successful group that (mostly) retained photosynthesis, so this is treated as somehow different when modeling ancient evolutionary events. We argue they should be treated equally: a loss is still a loss whether it preceded one modern species or a million of them.

The outright loss of the plastid organelle is very rare compared to the loss of photosynthesis, but this also happened in parallel in both

dinoflagellate and apicomplexan lineages. Plastids were lost multiple times in basal-branching apicomplexans[34] and at least once in dino-flagellates (Syndiniales)[14]. In all these cases, it is interesting to note how little evidence of the existence of a plastid is retained once the organelle has been lost. The idea that the endosymbiotic origin of plastids necessarily had an extensive and lasting genetic impact on their host genome has penetrated deeply into our thinking about organelle evolution, but the presence of a plastid in the ancestor of eleftherids and Syndiniales has left no impression on the genome of Syndiniales[15], and indeed the only reason we know a plastid ever existed is due to their phylogenetic position, not because of any genetic "footprint" of the organelle. This is also true of apicomplexan lineages that have lost their plastid entirely[34,53] and is further reinforced by the recent demonstration that the Picozoa are plastid-lacking members of the archaeplastids[55]. Therefore, perhaps our baseline expectation should be that when an organelle is lost, it leaves no genetic trace, making it very difficult, without clear phylogenetic evidence, to distinguish a cell where an organelle was lost from one that never had the organelle in the first place.

## Taxonomic summary
### Taxonomy
Eukaryota

SAR Burki et al. 2008, emend. Sar Adl et al. 2012

Alveolata Cavalier-Smith 1991

Dinoflagellata Bütschli 1885, emend. Fensome et al. 1993, emend. Adl et al. 2005

*Eleftheros* n. gen. Tikhonenkov, Hehenberger, and Keeling

**Etymology**: Generic epithet means 'free'. From Greek 'Eleftheria', freedom, liberty

**Diagnosis**: Naked and solitary eukaryovorous protist with two heterodynamic longitudinal and transverse flagella, large alveoli beneath the plasmalemma, bowling pin-shaped trichocysts, and rhoptry-like structures near the plasma membrane. Two basal bodies lie at an acute angle to each other or almost parallel. Large mitochondrion with tubular cristae.

**Taxonomic remark**: Free-living and fast-swimming cells outwardly resemble biflagellate dinospores of syndinian parasites (e.g. *Ichthyodinium, Amoebophrya*).

**Zoobank registration**: LSID for this publication: urn:lsid:zoobank.org:pub:CBA7B765-7996-4F42-8AAF-42460CE7F77E. LSID for the new genus: urn:lsid:zoobank.org:act:DD5CBE62-6058-4ADA-BDBD-7B3A486447E1.

**Type species**: *Eleftheros xomoi*

*Eleftheros xomoi* n. sp. Tikhonenkov, Hehenberger, and Keeling

**Diagnosis**: Cells 3.6-4.8 µm long, 3.0-4.1 wide, bean-shaped or oval with wider anterior end. Longitudinal flagellum strait, lies in the longitudinal groove and directed backward, 1.5 times longer than the cell. Transverse flagellum about 2 times as long as cell, undulates in the transverse groove, which descends down the cell and widens. Coming out of the transverse groove below the cell, transverse flagellum makes an undulating loop around the cell, while the tip of the flagellum is directed downward.

**Type figure**: Fig. 1a.

**Type material**: The specimen shown in Fig. 1a is the holotype (see International Code of Zoological Nomenclature, Art. 72.5.6, Declaration 45).

**Gene sequence**: The SSU rRNA gene sequence has the GenBank Accession Number OR427353.

**Type locality**: Coastal waters of Curacao

**Etymology**: Species epithet means 'phantom' in Aboriginal language of Curacao population.

**Zoobank registration**: LSID for this publication: urn:lsid:zoobank.org:pub:CBA7B765-7996-4F42-8AAF-42460CE7F77E. LSID for the

new species: urn:lsid:zoobank.org:act:1B42392D-9FE1-464F-B637-50761EFC7A08.

*Eleftheros karadeniz* n. sp. Tikhonenkov, Hehenberger, and Keeling

**Diagnosis**: Cells 4.2-5.5 µm long, roundish or wide-oval. Longitudinal flagellum strait, lies in the longitudinal groove and directed backward, 2 times longer than the cell. Transverse flagellum about 2.5 times as long as cell, undulates in the transverse groove.

**Type figure**: Fig. 1k.

**Type material**: The specimen shown in Fig. 1k is the holotype (see International Code of Zoological Nomenclature, Art. 72.5.6, Declaration 45).

**Gene sequence**: The SSU rRNA gene sequence has the GenBank Accession Number OR427351.

**Type locality**: Near shore bottom sediments in the Black Sea.

**Etymology**: The species epithet means 'Black Sea' in Turkish and Crimean Tatar languages.

**Zoobank registration**: LSID for this publication: urn:lsid:zoobank.org:pub:CBA7B765-7996-4F42-8AAF-42460CE7F77E. LSID for the new species: urn:lsid:zoobank.org:act:6264A9FC-BBD1-4D8D-B0E6-EB05C540D628.

## Methods
### Data reporting
No statistical methods were used to predetermine the sample size. The experiments were not randomized and the investigators were not blinded to allocation during experiments and outcome assessment.

### Cell isolation and culture establishment
Strains Colp-37 and Cur-11 (*Eleftheros xomoi* gen. et sp. nov.) were obtained from the surface of the brain coral *Colpophylia natans* Houttuyn, 1772 in coastal waters of Curaçao (Caribbean Sea) in April 2016 and 2018, respectively. Strain Colp-25 (*Eleftheros karadeniz* gen. et sp. nov.) was isolated from the near shore bottom sediments in the Black Sea near T.I. Vyazemsky Karadag Scientific Station, Crimea, May 2015. The samples were enriched with a suspension of *Aeromonas sobria* bacteria and examined on the third, sixth, and ninth day of incubation (25 °C, darkness) in accordance with methods described previously[56]. To obtain clonal cultures, the individual cells were transferred using a drawn-out glass micropipette into Petri dishes containing a clonal culture of the eukaryotic prey *Procryptobia sorokini* (strain B-69), which were grown in marine Schmalz–Pratt's medium at a final salinity of 20‰ using the bacterium *A. sobria* as food[57]. Strains perished after several months of cultivation.

The MALV-I infected *Polykrikos* cell was collected with a 20 µm mesh net towed at Heriot Bay near the Hakai Quadra Island Ecological Observatory in British Columbia, Canada (October 2022). The MALV-I infected *Warnowia* cell and host-less MALV-I were collected from near-shore tows at Piscadera Bay, Curaçao (February 2023), using the same net. Cells were isolated with a drawn-out glass micropipette and washed with (0.2 µm) filtered water before imaging and lysis.

*Psammosa pacifica* Psp strain was isolated from Boundary Bay, British Columbia, Canada (the type location of the original description) and maintained according to Okamoto et al. [35]. *Psammosa* sp. (strain Colp-34) was isolated from the near shore bottom sediments in the Kapsel Bay, Black Sea, Crimea (May 2016) and maintained following the abovementioned protocol.

### Light and electron microscopy
To observe living cells, an AxioScope A1 light microscope (Carl Zeiss, Jena, Germany) with DIC water immersion objective 63× and an inverted microscope Leica DM IL LED with DIC objectives 40× and 63× were used. Images were captured with a Sony α7 R camera.

For transmission electron microscopy (TEM), cells were centrifuged and fixed at 1 °C for 60 min in a cocktail of 0.6% glutaraldehyde and 2% $OsO_4$ (final concentration) prepared using a 0.1 M cacodylate buffer (pH 7.2). Fixed cells were dehydrated in alcohol and acetone series (30, 50, 70, 96, and 100%, 20 min in each step). Afterward, the cells were embedded in a mixture of Araldite and Epon (Fluka, 45345). Ultrathin sections (60 nm) were prepared with a Leica EM UC6 ultramicrotome (Leica Microsystems, Germany) and observed by using a JEM 1011 transmission electron microscope (JEOL, Japan).

## Preparation of libraries and sequencing

**RNA isolation and cDNA preparation.** Cells from clonal culture were collected by centrifugation (1000 × g, room temperature) onto the 0.8 μm membrane of a Vivaclear mini column (Sartorium Stedim Biotech Gmng, VK01P042). Total RNA was then extracted using a RNAqueous-Micro Kit (Invitrogen, AM1931) and reverse transcribed into cDNA using the Smart-Seq2 protocol[58], which uses poly-A selection to enrich mRNA. Additionally, cDNA of Cur-11 and Colp-37 were obtained from 20 single cells using the Smart-Seq2 protocol (cells were manually picked from culture using a drawn-out glass micropipette and transferred to a 0.2 mL thin-walled PCR tube containing 2 μL of cell lysis buffer −0.2% Triton X-100 and RNase inhibitor (Invitrogen)). Likewise, cDNA from the MALV-I associated cells and *Psammosa* spp. was prepared following the same protocol.

**Sequencing dataset assembly and decontamination.** The *Polykrikos-associated* MALV-I library was sequenced with an Illumina NextSeq 500 system using a Mid Output Flow Cell (2 x 150 bp reads), while the remaining two MALV-I associated cells were sequenced on the same system using a High Output Flow Cell (2 x 150 bp reads). *Eleftheros* and *Psammosa* libraries were sequenced on an Illumina MiSeq platform with read lengths of 2 × 300 bp (strains Colp-25, Colp-37, Psp, and Colp-34) and on an Illumina HiSeq 2500 machine, 2 × 125 bp reads (strain Cur-11). Sequence quality and adapter contamination of reads from transcriptomic datasets were assessed with FastQC v.0.10.1[59].

Reads of clone Colp-37 (culture and 20-cell preparations) and clone Colp-25 (culture) were merged with PEAR v.0.9.6[60] and resulting assembled as well as unassembled reads were separately trimmed with Trimmomatic[61] as implemented in Trinity v.2.0.6[62], removing Illumina adapters with ILLUMINACLIP, with a maximum of two mismatches, a palindrome clip threshold of 30 and a simple clip threshold of 10. Low-quality sequences were discarded, using a sliding window of 4 bp and a minimum trimmed length of 25 bp. Trimmed assembled and unassembled reads were combined into a single file and transcriptomes were assembled with Trinity, using the --single flag. Contaminating non-eukaryotic (bacterial, archaeal, and viral) and prey contigs were identified using BLASTn[63] queries of the NCBI nt database. Sequences that aligned with ≥100 nucleotides, had a query coverage of ≥80%, and were ≥90% identical to noneukaryotic entries were removed, while all sequences resulting in a kinetoplastid best hit were removed.

Reads of the later sampled clone Cur-11 (culture and 20-cell preparations) were trimmed and assembled without prior read merging in a single step, using the Trimmomatic plugin embedded in Trinity. Trimming parameters remained unchanged except for a simple clip threshold of 9. Due to results from initial analyses of clone Colp-37, the transcriptome was not subjected to any cleaning steps to retain putative bacterial horizontal gene transfers as well as sequences with bacterial best hits in nt that were identified as likely being eukaryotic in phylogenetic reconstructions in clone Colp-37. Instead, all genes of interest were subjected to phylogenetic analysis, as described below, to clarify the taxonomic identity of the gene.

Reads from the MALV-I-associated and psammosid libraries were trimmed using Cutadapt v3.2[64] before assembly with rnaSPAdes v3.15.1[65]. Contaminating sequences were removed using BLASTx[66] and BLASTn[63] searches against the NCBI nt and UniProt databases (E-value cut-off = 1×10⁻²⁵). Prey contigs, *Spumella elongata* for *Psammosa pacifica* and *Procryptobia sorokini* for *Psammosa* sp. were characterized using BLASTn[63] against prey transcriptome data and removed.

Prediction of protein coding regions in all assemblies was performed by TransDecoder v.5.0.2, including BLASTp[66] queries of the Swiss-Prot database (E-value cut-off = $1 \times 10^{-5}$).

## Phylogenomic dataset preparation and analysis

In addition to the taxa described above we added to or updated the following available transcriptome or genome data in an existing phylogenomic framework[67]. We added the re-assembled transcriptomes of several core dinoflagellates generated by the MMETSP project[68], several transcriptomes from the EukProt database[69], the transcriptomes of the dinoflagellates *Lepidodinium chlorophorum* (https://www.ncbi.nlm.nih.gov/bioproject/481676), TGD and MGD[70], *Abedinium*[71], *Amyloodinium ocellatum*[72] and additional taxa from Cooney et al. [73]. including *Spatulodinium, Kofoidinium* and *Fabadinium*. We updated the data for the MALVs *Amoebophrya ceratii*[74] and *Hematodinium* sp[14]., for *Oxyrrhis marina*[68], for the perkinsids *Perkinsus marinus* (https://protists.ensembl.org/Perkinsus_marinus_atcc_50983_gca_000006405/Info/Index), *Maranthos nigrum* and *Parvilucifera sinerae*[48], the apicomplexa *Ancora sagittate*[52], and for the chromerids *Vitrella brassicaformis* (https://cryptodb.org/cryptodb/app/record/dataset/NCBITAXON_1169540) and Symbiont X[52]. Transcriptomic datasets were subjected to TransDecoder coding region prediction to extract peptides for downstream analyses. Peptide sequences from predicted MALV SAGs described in Delmont et al. [28]. were also included.

All listed datasets were searched for 263 proteins to generate single-protein trees as described in Burki et al. [67]. In brief: BLASTp was used to identify homologs to the 263 genes in the new datasets. After a parsing step (E-value ≤ $1 \times 10^{-20}$), a maximum of four non-redundant hits was added to the initial 263-protein set. The expanded gene sets were aligned with MAFFT L-ins-i v.7.222[75] and trimmed automatically with trimAl v1.2[76], with a gap threshold of 80%. Single-protein ML phylogenies were reconstructed under the LG + G4 model using RAxML v.8.1.6[77] in combination with 100 rapid bootstraps and resulting trees were manually screened to flag paralogues and sequences derived from prey or other contamination. Cleaned protein sets were aligned and trimmed as above and taxa were selected upon concatenation with SCaFoS v.1.2.5[78] to select proteins sequences present in ≥60% of all taxa. To improve data presence, the concatenated sequences of clone Colp-37 derived from culture and 20-cell preparations were merged, as were the sequences for the two clone Cur-11 datasets. The final concatenated alignment included 75 taxa, 236 proteins and 63,267 amino acid sites. Clone Cur-11 was represented with 92% of proteins and 86% of sites, clone Colp-37 with 82% of proteins and 72% of sites and clone Colp-25 with 49% of proteins and 38% of sites. The host *Polykrikos* sp. was represented with 75% of proteins and 66% of sites and Ichthyodinida sp. 1 with 25% and 15% respectively. The host *Warnowia* sp. was represented with 61% of proteins and 39% of sites, and Ichthyodinida sp. 2 with 28% and 17%, respectively. Ichthyodinida sp. 3 was represented with 15% of proteins and 8% of sites *Psammosa pacifica* Psp was represented with 76% and 63% of sites, *Psammosa* sp. was represented with 75% of proteins and 65% of sites.

ML phylogenomic tree reconstruction was performed using IQ-TREE v. 1.6.5[79] using the C60 empirical mixture model in combination with the LG matrix, amino acid frequencies computed from the data, and four gamma categories for handling the rate heterogeneity across sites (LG + C60 + F + G4 model with 1000 UFBoot replicates[80]). Phylogenetic trees were visualized in R[81] with the ggtree[82] and treeio[83] packages.

Fast-evolving sites were estimated with IQ-TREE using the -wsr option, using the LG + C60 + F + G4 model. Sites were removed in increments of 5% of the original alignment length, up to 50%, and for

each subsample with a reduced number of sites, trees were reconstructed with the same model. The support for the sister relationship of eleftherids and MALVs was assessed at each removal increment. The monophyly of core dinoflagellates was tested as a control.

### Small subunit ribosomal RNA (SSU rRNA) phylogeny

The most complete sequences for SSU rRNA genes were identified from each eleftherid transcriptome with BLASTn, using a known dinoflagellate SSU query, whereas barrnap[84] was used to extract MALV-I SSU rRNA sequences prior to identification with BLAST. An alignment of all near full-length MALV sequences deposited in GenBank was curated to include all five previously described MALV groups in Guillou et al. [7]. The resulting SSU collection was aligned using MAFFT with the E-INS-I algorithm and inspected for misaligned and chimeric sequences. Shorter sequences obtained from surveys of environmental data (see below) were aligned using the addfragments flag. The final alignment was trimmed with trimAl (-gt 0.1, -st 0.001) before generating a ML tree in IQ-TREE with 1000 ultrafast bootstrap replicates[80], using the GTR + F + R7 model (and GTR + R + R6 for Supplementary Fig. 3), selected with ModelFinder[85]. *Oxyrrhis marina* and *Ellobiopsis chattonii* were omitted to minimize long branch attraction.

### Identification of putative plastid-targeted proteins

Putative plastid-targeted proteins were identified combining a BLASTp-based similarity search and a hidden Markov models (HMMs) based screen. Known dinoflagellate[39] or MALV[14] proteins involved in plastid metabolic pathways were used as queries in a BLASTp search against a comprehensive custom database containing representatives from most major eukaryotic groups (excluding the long-branching excavates and the data-poor group of Rhizaria) and RefSeq data from all bacterial phyla at NCBI (last accessed December 2017). The database was subjected to CD-HIT[86] clustering with a similarity threshold of 85% to reduce redundant sequences and paralogues, except for the data sets created in this study (clustered at 98%). The search results of the BLASTp step were parsed for hits with an E-value threshold $\leq 1 \times 10^{-25}$ and a query coverage of ≥30% to reduce the possibility of paralogs and extremely short sequences and at the same time recover possibly fragmented homologs. The number of bacterial hits was restrained to 20 hits per phylum (for FCB group, most classes of Proteobacteria, PVC group, Spirochetes, Actinobacteria, Cyanobacteria, and Firmicutes) or 10 per phylum (remaining bacterial phyla) as defined by NCBI taxonomy. In some cases (HemD and DapL) these numbers were expanded to 20 and 40, respectively, for a more representative bacterial sampling. In addition, protein data were used to search the Pfam-A database release 33.0 with hmmscan (HMMER3.1; hmmer.org), employing the manually curated Pfam gathering threshold. The results were queried, using a keyword search, for Pfam domains present in plastid-associated proteins, including proteins involved in metabolic pathways in Hehenberger et al. [39], photosynthesis, plastid import (TIC/TOC components) and Calvin cycle (RuBisCO). Candidates with domains of interest were used as BLASTp queries as described above, and parsed hits (query coverage of ≥50%) were combined with the recovered hits from the known proteins. After a deduplication step, sequences were aligned with MAFFT using the –auto option, trimmed using trimAl (-gt 0.8) and ML tree reconstructions were performed with FastTree v.2.1.7[87] using the default options in a preliminary analysis. The resulting phylogenies and underlying alignments were inspected manually to remove contaminant, divergent, and/or low-quality sequences. The cleaned, unaligned sequences were then subjected to filtering with PREQUAL[88] using the default options, followed by alignment with MAFFT G-INS-I using the VSM option (--unalignlevel 0.6). The alignments were subjected to Divvier[89] using the -mincol 4 and the -divvygap option before trimming with trimAl (-gt 0.01). Final trees were calculated with IQ-TREE, using the -mset option to restrict

model selection to LG for ModelFinder[85], while branch support was assessed with 1000 ultrafast bootstrap replicates[80].

We also searched for homologs to the 129 proteins predicted to localize to the apicoplast in *Toxoplasma*[42]. The predicted *Toxoplasma* proteins were used as queries in a BLASTp search against our custom database, using an initial E-value threshold of $\leq 1 \times 10^{-25}$ and a query coverage of ≥30% for parsing. All proteins recovering putative homologs were used in a preliminary tree reconstruction analysis as described above. After manual inspection of the phylogenies, potential plastid-targeted candidates were further investigated by using the recovered homologs as additional BLASTp queries and combining the resulting hits (query coverage of ≥50%) with the initial BLASTp output as described above. The BLASTp search using the *Toxoplasma* proteins was repeated with relaxed parameters (E-value threshold $\leq 1 \times 10^{-5}$) to recover additional candidates. All candidates investigated were also submitted to a web BLASTp search against nr to exclude the possibility of contamination not present in our database.

N-terminal extensions of putative plastid-targeted sequences were investigated in the respective protein alignments, visualized with AliView[90]. For easier recognition of such extensions, only the dinoflagellate sequences of the protein of interest plus the prokaryotic sequence with the highest sequence similarity were viewed. Sequences with N-terminal extensions relative to prokaryotic sequences were submitted to SignalP-3.0[91] and TMHMM 2.0[92] to predict putative signal peptides and transmembrane domains, respectively.

For proteins with known bacterial orthologs, bacterial naming conventions were applied, with the exception of the multifunctional enzyme Acetyl-CoA carboxylase (ACC) not present in bacteria and the enzyme 5-aminolevulinate synthase (ALAS), where eukaryotic conventions were applied.

### Identification of molecular characteristics of dinoflagellates

Dinoflagellate Viral NucleoProteins (DVNPs) were identified using a BLASTp search against our custom database using all described *Hematodinium* DVNP proteins[45] as well as by performing an hmmsearch (HMMER3.1; hmmer.org) against our database using the DVNP profile HMM downloaded from pfam.xfam.org, employing default thresholds. The two approaches recovered the same set of eleftherid DVNP candidates. No DVNP sequences were detected in psammosids or MALV-I. Only host DNVP sequences could be found in the *Polykrikos/*Ichthyodinida sp. 1 library.

We also searched for histone-like proteins (HLPs) using known representatives of the two known types of HLPs in dinoflagellates[36] as query, *Crypthecodinium cohnii* AAM97522.1 (HLPI; https://www.ncbi.nlm.nih.gov/protein/AAM97522.1) and *Noctiluca scintillans* ABV 22345.1 (HLPII; https://www.ncbi.nlm.nih.gov/protein/ABV22345) in a BLASTp search against the predicted peptides (E-value threshold $\leq 1 \times 10^{-25}$). Additionally, we performed an hmmsearch using a profile HMM constructed from a curated alignment of dinoflagellate and prokaryotic HLPs.

Dinoflagellate spliced leader (SL) sequences were identified using BLASTn searches (using the option -task blastn to allow for short input queries and to identify short matches) against all transcriptomes, using the canonical 21-nucleotide dinoflagellate SL sequence[47] as a query. The recovered 21-nucleotide sequence in eleftherids, differing in 2 nucleotides (5 A > T and 17 T > A) from the canonical sequence, was used as a search string to identify all eleftherids transcripts containing a full-length SL. To avoid SL sequences in host sequences, MALV-I peptides identified in single-gene trees were aligned against host/MALV-I transcripts with tBLASTn to obtain MALV-I only transcripts.

### Environmental distribution and abundance

To uncover the global distribution and abundance of eleftherids, we searched for sequences similar to the eleftherid SSU rRNA gene among published environmental SSU rRNA gene amplicon studies.

Full-length eleftherid SSU rRNA genes and their V4 and V9 hypervariable regions were used as queries in BLASTn searches (E-value threshold ≤1×10⁻¹⁰) against the complete Tara Ocean database available on Ocean Gene Atlas[18]. We recovered no sequences with ≥95% identity and a query cover of ≥90%, neither when using full-length nor hypervariable regions of eleftherid SSU rRNA gene queries.

Additionally, we searched a custom environmental sequence database containing data from 26 environmental amplicon studies with a focus on marine sediment, also including the large sequencing projects BioMarKs[4] and Malaspina[19]. NCBI SRA files of amplicon data were downloaded using fastq-dump from sratoolkit 2.10.8. Raw sequence data were processed with MICrobial Community Analysis, using micca merge or mergepairs (-l 100 -d 30) for single-end or paired-end data respectively[93], and then concatenated and converted them into a BLAST database composed of 113,203,549 sequences with a total of 25,261,490,053 bases. Alternatively, paired-end data were processed in Qiime2[94] with the DADA2[95] denoising algorithm before creation of the BLAST database. This database was searched using full-length eleftherid SSU rRNA gene sequences as well as their V4 and V9 hypervariable regions in BLASTn searches (E-value threshold ≤1×10⁻²⁵). In addition to the environmental database, sequences similar to the eleftherid 18 S rRNA gene were searched using BLASTn against a V9 amplicon dataset collected from sandy beaches off the central coast of British Columbia (the European Nucleotide Archive project PRJEB14727). A total of 73 unique sequences with ≥97% identity to and ≥90% coverage of the eleftherid V9 and V4 hypervariable regions resulted from both search approaches. Full-length and hypervariable regions V9 and V4 of *P. pacifica* were used as queries against the environmental database which resulted in a total of 4 unique sequences with ≥97% identity to and ≥90% coverage to psammosids. These sequences were then clustered at 99% identity using CD-HIT prior to being added to the SSU rRNA gene phylogeny to confirm their phylogenetic position as described above. Sequences less than 150 bp were removed from the final phylogeny. Sample coordinates corresponding to BLAST hits in the custom database were extracted and plotted in R[81] with the rnaturalearth[96] package.

### Reporting summary

Further information on research design is available in the Nature Portfolio Reporting Summary linked to this article.

## Data availability

Raw transcriptome reads have been deposited in the GenBank Sequence Read Archive (SRA) database under the accessions SRR25604407−SRR25604420. SSU rRNA gene sequences retrieved from the transcriptomes have also been deposited in GenBank under the accessions OR427349-OR427355. All sequence data is linked to the BioProject accession code PRJNA1003956. Assembled transcriptomes, along with individual gene alignments, concatenated and trimmed alignments, and ML tree files for the phylogenomic dataset are available at Figshare (https://doi.org/10.6084/m9.figshare.24293662). The untrimmed and trimmed alignments, alignments depicting N-terminal extensions and tree files in nexus and pdf format for plastid-associated and other proteins of interest are available at Figshare (https://doi.org/10.6084/m9.figshare.19351700). The genus *Eleftheros* (urn:lsid:zoobank.org:act:DD5CBE62-6058-4ADA-BDBD-7B3A486447E1) and species *Eleftheros xomoi* (urn:lsid:zoobank.org:act:1B42392D-9FE1-464F-B637-50761EFC7A08) and *Eleftheros karadeniz* (urn:lsid:zoobank.org:act:6264A9FC-BBD1-4D8D-B0E6-EB05C540D628) have been registered with the Zoobank database (http://zoobank.org/). Zoobank Registration: LSID for this publication: urn:lsid:zoobank.org:pub:CBA7B765-7996-4F42-8AAF-42460CE7F77E.

## Code availability

All unpublished code is available upon request from the corresponding authors.

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

## Acknowledgements

We thank A.P. Mylnikov and T.G. Simdyanov for help with sample collection, fixation, and interpretation of transmission electron microscopy images. This research was supported by grants from the Hakai Institute (P.J.K), the Gordon and Betty Moore Foundation (https://doi.org/10.37807/GBMF9201, P.J.K.), the Natural Sciences and Engineering Research Council of Canada (Grant Number 2019-03994, P.J.K), the Czech Academy of Sciences (Grant Number LQ200962204, E.H.) the Russian Foundation for Basic Research (Grant Number 20-04-00583, D.V.T.), Tyumen Oblast Government, as part of the West-Siberian Interregional Science and Education Center's project No. 89-DON (2) and carried out within the framework of State Assignment no. 121051100102-2.

## Author contributions

C.C.H., E.H., D.V.T., and P.J.K. designed the study. D.V.T. is isolated and cultured eleftherid cells. C.C.H. and E.C.C. isolated MALV-I. V.K.L.J.-R and N.O. isolated and cultured psammosid cells. C.C.H., E.H., D.V.T., V.K.L.J.-R., and E.C.C. generated material for sequencing. D.V.T. performed microscopy experiments. C.C.H., E.H., E.C.C., and N.A.T.I. performed phylogenomic analyses. C.C.H. and E.C.C. performed

phylogenetic analysis of the SSU rRNA genes. C.C.H. analyzed published SAG data. V.K.L.J.-R. performed the environmental distribution and abundance analysis. C.C.H., E.H., and V.K.L.J.-R. performed transcriptomic analyses and phylogenetic analysis of plastid and other proteins. C.C.H., E.H., and P.J.K. wrote the manuscript with input from all authors.

## Competing interests

The authors declare no competing interests.
