## [Peer Review File · Nature Communications]

Multiple parallel origins of parasitic Marine AlveolatesEditorial Note: This manuscript has been previously reviewed at another journal that is not operating a transparent peer review scheme. This document only contains reviewer comments and rebuttal letters for versions considered at Nature Communications.

Reviewers' Comments:

Reviewer #3:

Remarks to the Author:

This is a third version of the manuscript I am reviewing, so I refrain from commenting in detail on the general significance of the study and refer to my previous reports, especially the one to the second version (initially submitted to Nature), which already had a scope very similar to the present version. Hence, briefly, this is a very important study on a critical sector of eukaryote phylogenetic diversity and makes a major leap in our understanding of protist phylogeny and diversity as well as evolution of key traits (plastids, parasitism). The new version has been further boosted by additional data, making it ever stronger than the previous version. Critically, the authors have carefully addressed all my major concerns with the previous version, and while there still might be conceptual details to argue about, I do not find them significant enough to voice them here. I wanted to thank the authors for the exchange of arguments in their rebuttal letters, I enjoyed our conversation very much and I also learned something from it. This is how peer review should look, I believe. That said, I have noticed only a few minor formal issues (listed below) that in principle can be addressed in proofs, so I fully support publication of this highly significant study that will for sure be read and appreciated by a wide community of biologists.

Minor issues:

Line 209-210: "HemH and HemF are truncated at the N-terminus but cluster with plastid-targeting dinoflagellates" – XXXX; similarly "plastid-targeted dinoflagellates" on line 212

Line 210-211: "while HemE, also contains N-terminal targeting information" – I think the comma should not be used in this sentence

Reference #23: perhaps the editors of the book containing the chapter cited in the reference should be mentioned. Furthermore, note that the reference #29 is the same as the reference #23, so fix the redundancy, please.

Reference #32: again I guess editors of the book cited should be indicated to provide a full reference.

Legend to Fig. 2: I believe that the sentence starting "Support for core dinoflagellates" is possibly intended as a remark to panel b instead of panel a, so please check and move to the description of panel b if needed.

Reviewer #3 (Remarks to the Author):

This is a third version of the manuscript I am reviewing, so I refrain from commenting in detail on the general significance of the study and refer to my previous reports, especially the one to the second version (initially submitted to Nature), which already had a scope very similar to the present version. Hence, briefly, this is a very important study on a critical sector of eukaryote phylogenetic diversity and makes a major leap in our understanding of protist phylogeny and diversity as well as evolution of key traits (plastids, parasitism). The new version has been further boosted by additional data, making it ever stronger than the previous version. Critically, the authors have carefully addressed all my major concerns with the previous version, and while there still might be conceptual details to argue about, I do not find them significant enough to voice them here. I wanted to thank the authors for the exchange of arguments in their rebuttal letters, I enjoyed our conversation very much and I also learned something from it. This is how peer review should look, I believe. That said, I have noticed only a few minor formal issues (listed below) that in principle can be addressed in proofs, so I fully support publication of this highly significant study that will for sure be read and appreciated by a wide community of biologists.

We would like to thank the reviewer for their kind words and dedication to improve our manuscript. We have enjoyed the discussion and are glad to see that we have addressed all major concerns. This process has led to thoughtful additions to our paper and for that we are grateful.

Minor issues:

Line 209-210: “HemH and HemF are truncated at the N-terminus but cluster with plastid-targeting dinoflagellates” – XXXX; similarly “plastid-targeted dinoflagellates” on line 212
The language has been standardised throughout – genes/proteins are referred to as “plastid-targeted” but the dinoflagellates themselves as “plastid-targeting”.

Line 210-211: “while HemE, also contains N-terminal targeting information” – I think the comma should not be used in this sentence
We have removed the comma.

Reference #23: perhaps the editors of the book containing the chapter cited in the reference should be mentioned. Furthermore, note that the reference #29 is the same as the reference #23, so fix the redundancy, please.
All reference numbers have been updated reflecting rearrangement of manuscript sections to comply with the author checklist. Editor names have been added.

Reference #32: again I guess editors of the book cited should be indicated to provide a full reference.
Editor names have been added.

Legend to Fig. 2: I believe that the sentence starting “Support for core dinoflagellates” is possibly intended as a remark to panel b instead of panel a, so please check and move to the description of panel b if needed.
The reviewer is correct. We have moved the sentence.